# Validation of a Deep Learning Model for Detecting Chest Pathologies from Digital Chest Radiographs

**DOI:** 10.3390/diagnostics13030557

**Published:** 2023-02-02

**Authors:** Pranav Ajmera, Prashant Onkar, Sanjay Desai, Richa Pant, Jitesh Seth, Tanveer Gupte, Viraj Kulkarni, Amit Kharat, Nandini Passi, Sanjay Khaladkar, V. M. Kulkarni

**Affiliations:** 1Dr. D.Y. Patil Hospital, D.Y. Patil University, Pune 411018, India; 2Lata Mangeshkar Hospital, Nagpur 440012, India; 3Deenanath Mangeshkar Hospital and Research Centre, Pune 411004, India; 4DeepTek Medical Imaging Pvt Ltd., Pune 411045, India; 5Post Graduate Institute of Medical Education and Research, Chandigarh 160012, India

**Keywords:** chest X-ray, AI, multi reader multi case (MRMC), AUROC, region of interest (ROI), lungs, pleura, cardiac

## Abstract

**Purpose:** Manual interpretation of chest radiographs is a challenging task and is prone to errors. An automated system capable of categorizing chest radiographs based on the pathologies identified could aid in the timely and efficient diagnosis of chest pathologies. **Method:** For this retrospective study, 4476 chest radiographs were collected between January and April 2021 from two tertiary care hospitals. Three expert radiologists established the ground truth, and all radiographs were analyzed using a deep-learning AI model to detect suspicious ROIs in the lungs, pleura, and cardiac regions. Three test readers (different from the radiologists who established the ground truth) independently reviewed all radiographs in two sessions (unaided and AI-aided mode) with a washout period of one month. **Results:** The model demonstrated an aggregate AUROC of 91.2% and a sensitivity of 88.4% in detecting suspicious ROIs in the lungs, pleura, and cardiac regions. These results outperform unaided human readers, who achieved an aggregate AUROC of 84.2% and sensitivity of 74.5% for the same task. When using AI, the aided readers obtained an aggregate AUROC of 87.9% and a sensitivity of 85.1%. The average time taken by the test readers to read a chest radiograph decreased by 21% (*p* < 0.01) when using AI. **Conclusion:** The model outperformed all three human readers and demonstrated high AUROC and sensitivity across two independent datasets. When compared to unaided interpretations, AI-aided interpretations were associated with significant improvements in reader performance and chest radiograph interpretation time.

## 1. Introduction

Pulmonary and cardiothoracic disorders are among the leading causes of morbidity and mortality worldwide [1]. Chest radiography is an economical and widely used diagnostic tool for assessing the lungs, airways, pulmonary vessels, chest wall, heart, pleura, and mediastinum [2]. Since modern digital radiography (DR) machines are quite affordable, chest radiography is widely used in the detection and diagnosis of multiple chest abnormalities such as consolidations, opacities, cavitations, blunted costophrenic angles, infiltrates, cardiomegaly, nodules, etc. [3]. Each chest X-ray (CXR) image contains a huge amount of anatomical and pathological information packed into a single projection, potentially making disease detection and interpretation difficult [4]. The correct interpretation of information is always a major challenge for medical practitioners. Pathologies such as lung nodules or consolidation may be obscured by superimposed dense structures (for example, bones) or by poor tissue contrast between adjacent anatomical structures [4]. Moreover, low contrast between the lesion and the surrounding tissue, and an overlap of the lesion with ribs or large pulmonary vessels make the detection of the disease even more challenging [3]. As a result, examination of chest pathologies from a CXR may result in some missed detection. The difficulty of missed radiological findings is exacerbated by the increase in the number of examinations, which in turn is happening much faster than the rise in the number of qualified radiologists [5]. Due to heavy workloads in healthcare facilities and a scarcity of experienced radiologists in developing nations, timely reporting of every image is not always possible [6]. Therefore, computer-aided detection (CAD) systems are gaining popularity and acceptance because they can assist radiologists in detecting suspicious lesions that would otherwise be easily missed, thereby improving detection accuracy [7]. An automated system can also help control the variability among radiologists and advise them on further interpretation of abnormal cases [8].

The phenomenal success of deep learning techniques such as convolutional neural networks (CNNs) for image classification tasks renders these algorithms a potential candidate for automated CXR analysis [9]. Several studies have been conducted for the diagnosis of chest diseases using artificial intelligence methodologies. Deep learning models have been employed for the diagnosis of specific chest abnormalities such as lung nodules [10,11], COVID-19 [12,13], pulmonary tuberculosis classification [14], cardiomegaly detection [15,16], etc. However, these algorithms are disease/condition specific. Most of the algorithms tested in clinical settings have limited utility, as a relatively large number of diseases and anomalies exist in real-world clinical practice.

For a CAD system to have better clinical utility, it should be able to detect a variety of abnormalities from a chest radiograph. These include thoracic diseases, which constitute most of the abnormalities found on chest radiographs. Recent research studies utilized different AI models to detect multiple abnormalities from chest radiographs. Most of these studies, however, validated the performance of their standalone AI models rather than comparing them to human readers. Table 1 provides an overview of the literature by detailing the validation dataset, problem statement, methodology, results, advantages, and limitations for each referenced article. Out of all the studies listed in Table 1, only Hwang et al. compared the performance of their model with human readers. However, the abnormal chest radiographs in their study were indicative of only one target disease, which does not represent the real-world clinical situation. In this article, we evaluate an AI system for the detection of multiple chest pathologies seen on chest radiographs and compare its performance with experienced human readers on clinical data from two different hospitals. We also aim to compare the performance of radiologists assisted by the CAD system to that of radiologists without assistance. The results indicate that the AI model outperformed experienced human readers in accurately identifying abnormal chest radiographs and classifying them into one or more of the three different categories: lungs, pleura, and cardiac. Readers aided by the AI system showed improvement in AUROCs and sensitivities, and took less time analyzing the radiograph and identifying the abnormality.

## 2. Materials and Methods

This study was approved by the institutional review boards (IRBs) of both the participating hospitals (Hospital A and Hospital B). Because of the retrospective nature of the study, the need for separate patient consent was waived by the IRB of both institutions. The external validation of the AI model was performed using data collected between January 2021 to April 2021 from these two hospitals. A total of 4763 chest radiographs were used for external evaluation.

### 2.1. Data Collection

To acquire data, the chest radiographs were downloaded from Picture and Archival Communication System (PACS) in a Digital Imaging and Communication in Medicine (DICOM) format. The data were downloaded in an anonymized format and in compliance with the Health Information Portability and Accountability Act (HIPAA).

Chest radiographs with both PA and AP views were included in the study. Radiographs acquired in an oblique orientation or processed with significant artifacts were excluded from the study. The inclusion and exclusion criteria used for the selection of chest radiographs are presented in Figure 1. The chest radiographs were acquired on multiple machines of different milliamperes (mAs). These included multiple computed radiography (CR) systems, such as Siemens 500 mA Heliophos-D, Siemens 100 mA Genius-100R, Siemens 300 mA Multiphos-15R; and a 600 mA digital radiography (DR) system, the Siemens Multiselect DR. Some of the radiographs were acquired on the Siemens 100 mA and Allengers 100 mA portable devices. The plate sizes used for the CR system were the standard 14 × 17 inch for adults. For the DR system, a Siemens detector plate was used.

### 2.2. Establishing Ground Truth

To establish the ground truth, chest radiographs were classified into lungs, pleura, and cardiac categories by three board-certified radiologists with a combined experience of 21+ years. ‘Lungs’ included pathologies such as tuberculosis, atelectasis, fibrosis, COVID-19, mass, nodules, opacity, opaque hemithorax, etc.; ‘pleura’ included pathologies such as pneumothorax, pleural thickening, pleural effusion, etc.; and ‘cardiac’ included pathologies that result in enlargement in the size of heart, such as cardiomegaly, pericardial effusion, etc. Normal radiographs and radiographs with medical devices (e.g., chest tubes, endotracheal tubes, lines, pacemakers, etc.) or chest abnormalities with ROIs in none of the above categories were binned in a separate category. The ground truth label for the presence or absence of ROI for each category was defined as the majority opinion of 2 out of the 3 readers.

### 2.3. AI Model

All 4476 chest radiographs were de-identified and processed with DeepTek Augmento, a cloud-based AI-powered PACS platform. Augmento [22] can identify multiple abnormalities from different categories and is currently used by more than 150 hospitals and imaging centers worldwide. It examines adult digital chest radiographs for various abnormalities and identifies, categorizes, and highlights suspicious regions of interest (ROIs) using the deployed AI models. The AI models were trained on over 1.5 million chest radiographs manually annotated by expert board-certified radiologists. The models use a series of convolutional neural networks (CNNs) to identify different pathologies on adult frontal chest radiographs. The processing of chest radiographs involves the following steps. Each radiograph is resized to a fixed resolution and normalized to standardize the acquisition process. The CNN parameters are optimized using appropriate loss functions and optimizers. Optimal thresholds are determined using a proprietary DeepTek algorithm. These thresholds were assessed using a validation set that had not been used for training the models. The radiographs used in this study were not augmented or processed further. Augmento is an ensemble of more than 16 models, each of which is used to detect specific abnormalities in the adult chest radiograph. It takes less than 30 s to process and report each radiograph. Readers can read and annotate scans on the Augmento platform. The platform also provides AI predictions for assistance and generates the report based on the annotations made and accepted by the readers. Figure 2 presents a screenshot of the Augmento platform.

### 2.4. Multi Reader Multi Case (MRMC) Study

An MRMC study was conducted to evaluate whether the AI aid can improve readers’ diagnostic performance in identifying chest abnormalities. A panel of three readers (R1, R2, and R3) with 2, 11, and 3 years of experience, respectively was established. For the MRMC study, external validation datasets from two hospitals were used. The radiologists who established the ground truth for the entire dataset were excluded from participating in the study. The study was conducted in two sessions. In session 1 (unaided session), readers independently assessed every CXR without the assistance of the AI to categorize the suspicious ROIs present in the chest radiographs into three classifications: lungs, pleura, and cardiac. After a washout period of one month to avoid memory bias, readers reevaluated each CXR with the assistance of AI in session 2 (aided session). The evaluation workflow for the unaided and aided readings was identical except that, during the aided reading session, readers could see the AI-suggested labels and bounding boxes over suspicious ROIs.

### 2.5. Statistical Analysis

To compare the AUROCs of readers between session 1 and session 2, the fixed readers random cases (FRRC) paradigm of the OR [23] method was used. The analysis was conducted in R (version 4.2.1, Vienna, Austria) using the RJafroc library (version 2.1.1). To compare the sensitivity and specificity of readers, a one-tailed Wilcoxon test was performed on ten independent samples of reader annotations. To compare the average time taken by the readers to read one radiograph between two sessions, a one-tailed Wilcoxon test was performed. A *p*-value of less than 0.05 was considered statistically significant.

## 3. Results

### 3.1. Data Characteristics

A total of 4476 chest radiographs were used to evaluate the performance of the model on two independent test sets. The average age of the patients was 41.1 ± 19.6 years in the dataset from hospital A and 36.6 ± 18.6 years in the dataset from hospital B. Out of 4476 frontal chest radiographs, 59.5% were from male patients and 40.4% were from female patients. The distribution of scans across lungs, pleura, and cardiac categories is represented in Table 2.

### 3.2. Standalone Performance of the AI Model

The performance of the AI model on the external dataset revealed an aggregate AUROC of 91% and 91.9% on data from Hospital A and Hospital B, respectively. The model achieved an aggregate sensitivity of 87.6% and 92%, and a specificity of 88.5% and 88.7% on data from hospitals A and B, respectively. The performance of the model on the dataset from hospital A demonstrated an AUROC of 88.6% for lungs, 86.7% for pleura, and 91.9% for cardiac. On the dataset from hospital B, the model demonstrated an AUROC of 90.2% for lungs, 87.1%, for pleura, and 85.5% for cardiac (Figure 3). Over the entire dataset, the model achieved an aggregate sensitivity of 85.5%, 77.9%, and 85.2% in detecting suspicious ROIs in the lungs, pleura, and cardiac, respectively. Similarly, the aggregate specificity in detecting suspicious ROIs in the lungs, pleura, and cardiac was 87.8%, 93.8%, and 92.7%, respectively.

The category-wise AUC, sensitivity, specificity, accuracy, F1 score, and NPV of the AI model on both datasets are presented in Table 3. The outputs of the model were visualized as bounding boxes enclosing the suspicious ROIs (Figure 4).

### 3.3. Comparison between the AI Model and Human Readers

The standalone AI model had an aggregate AUROC of 91.2% and a sensitivity of 88.4% across both hospitals. In session 1 (unaided session) of the MRMC study, the aggregate AUROC and sensitivity for human readers across both hospitals were 84.2% and 74.5%, respectively. The aggregate AUROC and sensitivity of the AI model were significantly higher (** *p* < 0.01) than the aggregate sensitivity and specificity of all 3 readers across the two hospital datasets. However, the aggregate specificity of the model was lower than the specificity of the human readers.

### 3.4. Comparison between Human Readers in Unaided and Aided Sessions

In session 2 of the MRMC study, the aggregate AUROC of test readers improved from 84.2% in the unaided session to 87.9% in the aided session across both hospitals. AI assistance significantly improved the aggregate sensitivity of test readers from 74.5% to 85.1% across both hospitals. While there was a significant improvement (** *p* < 0.01) in the aggregate AUROC and sensitivity of all three readers across different hospitals, there was no significant improvement in aggregate specificity values, as they remained consistently high for the readers in both sessions. Table 4 compares the AUROC, sensitivity, and specificity of the unaided and aided readers in the individual hospital datasets.

The aggregate performances of the unaided and aided readers (RI, R2, and R3) across all categories and hospital datasets are tabulated in Appendix A Appendix A. The aggregate sensitivity and specificity of different readers (R1, R2, and R3) in unaided and aided reading sessions, using the consensus of three board-certified radiologists as a ground truth reference standard, are shown in Figure 5.

### 3.5. Reduction in False-Negative Findings

AI assistance helped the test readers identify true positive cases and reduce false-negative findings. In some cases, unaided readers missed the pathology, but AI detected it. In such cases, readers could identify pathologies only with the assistance of AI. Figure 6 depicts the representative images from the MRMC study.

### 3.6. Interpretation Time for Each Radiograph

To test the effect of AI aid on the interpretation time of chest radiographs, the time spent by each reader on each radiograph in both the unaided and aided reading sessions was recorded. The mean chest radiograph interpretation time of the three readers decreased in the AI-aided reading session compared with the unaided reading session (time per chest radiograph: 13.43 ± 24.92 s vs. 10.61 ± 33.66 s; *p* < 0.001) (Appendix A Appendix A).

## 4. Discussion

In this study, we validated an AI model to classify chest radiographs with abnormal findings indicative of pathologies pertaining to the lungs, pleura, and cardiac regions on two different hospital datasets. The standalone performance of the AI model was significantly better than the performance recorded by the human readers in both unaided and AI-aided sessions. We also demonstrated significant improvement in reader performance (AUC and sensitivity) and productivity (reduction in time to report a radiograph) with AI assistance.

Recent studies have demonstrated the use of deep convolutional neural networks to identify abnormal CXRs for automated prioritization of studies for quick review and reporting [19,20]. Annaruma et al. used their AI system for automated triaging of adult chest radiographs based on the urgency of imaging appearances. Although their AI system was able to interpret and group the chest radiographs based on the prioritization categories, the AI performance could appear exaggerated if the scan was added to the correct priority class for the wrong reasons. Dunnmon et al. demonstrated the high diagnostic performance of CNNs trained with a modestly sized collection of CXRs in identifying normal and abnormal radiographs [20]. Although their training set was large (containing 216,431 frontal chest radiographs), they evaluated their CNNs on a held-out dataset of 533 images. Nguyen et al. measured the performance of their AI system on 6285 chest radiographs extracted from the Hospital Information System (HIS) in a prospective study [21]. Their system achieved an accuracy of 79.6%, a sensitivity of 68.6%, and a specificity of 83.9% on the prospective hospital dataset. However, the study did not assess the effect of the AI system on reader performance, and only provided a broad evaluation of the system for classifying a chest radiograph into normal or abnormal. Albahli et al. used ResNet-152 architecture trained on six disease classes and obtained an accuracy of 83% [22]. The model used in our study obtained an accuracy of 88.5% in classifying diseases into four categories suggestive of multiple disease conditions. Hwang et al. validated their AI algorithm on five external test sets containing a total of 1015 chest radiographs [23]. Although their model outperformed human readers and demonstrated consistent and excellent performance on all five external datasets, it covered only 4 major thoracic disease categories (pulmonary malignant neoplasm, active tuberculosis, pneumonia, and pneumothorax). Additionally, each abnormal chest radiograph in their external validation data sets represented only 1 target disease, which does not replicate a real-world situation.

The chest radiographs used in our study were closely representative of real-world clinical practice, as we did not segregate the chest radiographs based on the presence of only a single target condition. The chest radiographs were obtained from two different hospital settings and each abnormal radiograph was representative of one or multiple chest conditions. The AI model utilized in our study classified chest radiographs into three categories, i.e., lungs, pleura, and cardiac. The strength of this approach is that the identified ROIs could be suggestive of different conditions/pathologies pertaining to these categories. This can help human readers identify the categories of the suspected abnormality and define the appropriate prognosis. According to our study, the AI model showed promising results in identifying and categorizing chest abnormalities. The model was highly specific (with an aggregate specificity on the entire dataset of 88.5%) in identifying suspicious ROIs in the lungs, pleura, and cardiac regions. The model demonstrated an aggregate AUC of 91.2% and a sensitivity of 88.4% and outperformed unaided human readers, who achieved an aggregate AUC of 84.2% and a sensitivity of 74.5% across all datasets. The high aggregate NPV (96.3%) of the model demonstrates its utility in finding and localizing multiple abnormalities in CXRs. The consistently high performance of the model on both datasets without the interference of human readers suggests that it has the potential to be used as a standalone tool in clinical settings. Additionally, the AI assistance significantly improved the aggregate AUROC (from 84.2% to 87.9%) and sensitivity (from 74.5% to 85.1%) of test readers across both hospital datasets. The improvement in reader sensitivity implies a reduction in false negative findings and fewer disease cases missed. This is clinically important because false negative findings lead to missed diagnoses, thereby increasing the disease burden. The AI aid used in our study demonstrated a positive effect on CXR reporting time. When using the AI aid, the average time taken by human readers to read a chest radiograph decreased significantly by 21%. The significant reduction in the time required to read a chest radiograph signifies the utility of the AI aid in reducing delays in reporting. AI also assisted readers in identifying pathologies that they would have otherwise missed. This helps radiologists detect complex pathologies and prioritize images with positive findings in the read queue.

Our study had some limitations. First, the specificity of the AI model was low when compared to human readers. However, in an actual clinical setting, sensitivity is a more meaningful metric to measure model performance. Although identifying both true positives and false negatives is important, missing a true positive case may have greater consequences for patients’ health. Second, we reported only the image-level performance of the model and readers and did not evaluate the location-level performance. Future work will include the evaluation of localization performance for more accurate results. Third, we designed the study to include suspicious ROIs present only in the lungs, pleura, and cardiac regions. The suspicious ROIs present in categories other than lungs, pleura, and cardiac were binned in one separate category. Including abnormalities of other regions (such as mediastinum, hardware, bones, etc.) in different categories might not be beneficial at this point as it may result in many false positive classifications, thus hampering the clinical utility of the model. We believe that the AI model used in this study can detect substantial proportions of lung and cardiothoracic diseases in clinical practice. Fourth, the time taken by readers to report a chest radiograph was measured using the difference between the study opening time and the study submission time in both the unaided and aided sessions. This is not representative of turnaround times in clinical settings, which include more steps. However, the average turnaround times measured in this study provide a general idea of the utility of the model in reducing delays in reporting. Fifth, this was a dual-centered retrospective study. Although the AI model used in the study is generalizable on both hospital datasets, further research would be required to establish the generalizability of the model across different geographies.

## 5. Conclusions

In conclusion, we demonstrated the feasibility of an AI model in classifying radiographs into different categories of chest abnormalities. The high performance of the deep learning model in classifying abnormal chest radiographs, outperforming even human readers, suggests its potential for standalone use in clinical settings. It may also improve radiology workflow by aiding human readers in faster and more efficient diagnoses of chest conditions. The study showed promising results for future clinical implementation.

## Figures and Tables

**Figure 1 diagnostics-13-00557-f001:**
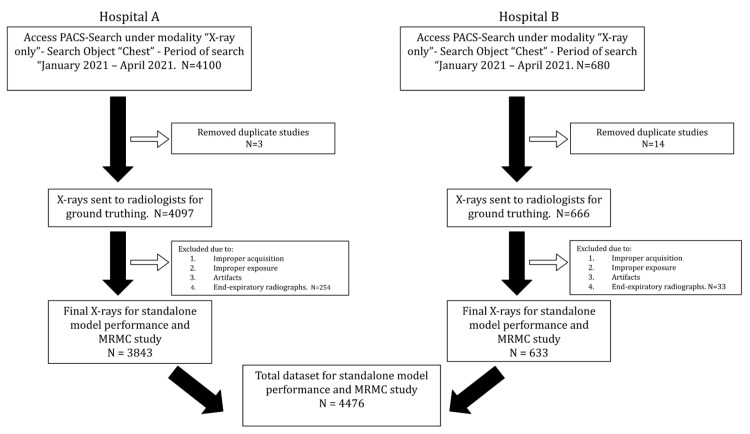
Flowchart illustrating inclusion and exclusion criteria used to select chest radiographs for standalone model performance and multi-reader multi-case (MRMC) study.

**Figure 2 diagnostics-13-00557-f002:**
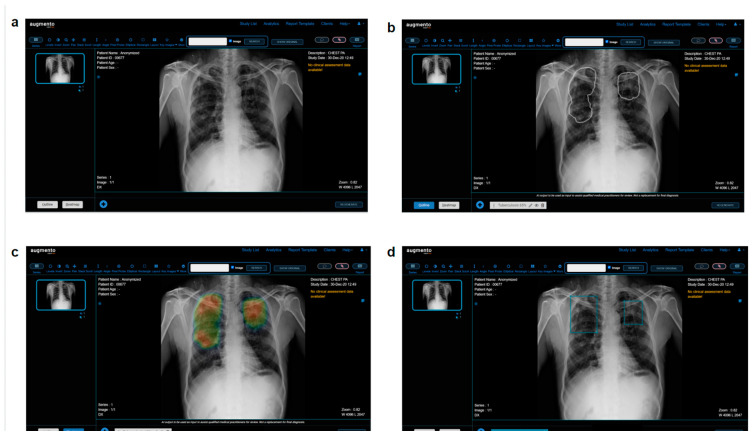
(**a**) Screenshot of Augmento, as seen by human readers. The readers can check the AI predictions in the (**b**) outline or (**c**) heatmap view and annotate the scans using (**d**) bounding boxes. Once the annotations are complete, a radiology report is generated.

**Figure 3 diagnostics-13-00557-f003:**
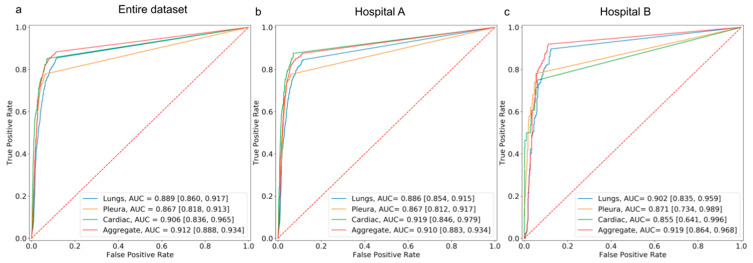
AUROC curves depicting the performance of the standalone AI model in detecting suspicious ROIs in the lungs, cardiac, and pleura categories on (**a**) the entire hospital dataset, (**b**) the dataset from Hospital A, and (**c**) the dataset from Hospital B.

**Figure 4 diagnostics-13-00557-f004:**
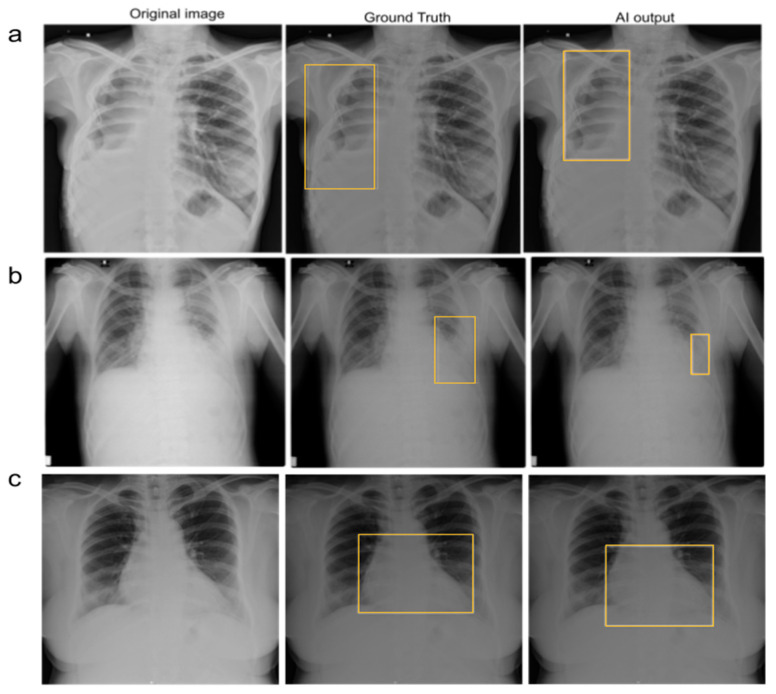
(Left to right) Original chest radiograph, with a ground truth bounding box indicating the presence of suspicious ROI, and an AI-generated bounding box indicating the presence of suspicious ROI in (**a**) lungs, (**b**) pleura, and (**c**) cardiac regions.

**Figure 5 diagnostics-13-00557-f005:**
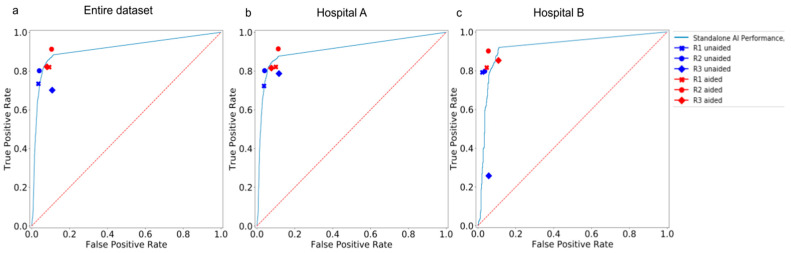
AUROC curves depicting the performance of standalone AI, unaided readers, and aided readers on (**a**) the entire hospital dataset, (**b**) the dataset from Hospital A, and (**c**) the dataset from Hospital B.

**Figure 6 diagnostics-13-00557-f006:**
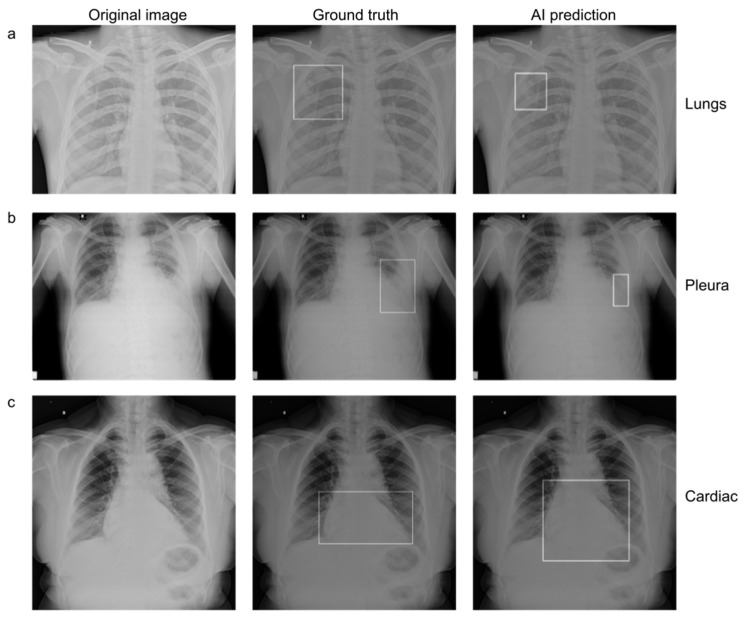
Examples of chest radiographs with suspicious ROIs in the (**a**) lungs, (**b**) pleura, and (**c**) cardiac categories. All three test readers missed these suspicious ROIs in the unaided session. The AI model and ground truth readers, however, predicted the suspicious ROIs as shown with bounding boxes. All three test readers identified suspicious ROIs in each category correctly when aided by AI.

**Table 1 diagnostics-13-00557-t001:** A literature review of research articles investigating the role of AI in detecting pathologies from chest radiographs.

Study	Size of Test/Validation Dataset	Problem Statement	Method	Results	Advantages	Limitations
Annarumma et al., 2019 [17]	3229 institutional adult chest radiographs	Developed and tested an AI model, based on deep CNNs, for automated triaging of adult chest radiographs based on the urgency of imaging appearances	Ensemble of two deep CNNs	Sensitivity of 71%, specificity of 95%, PPV of 73%, and NPV of 94%	The AI model was able to interpret and prioritize chest radiographs based on critical or urgent findings	False Negatives could result in misinterpretation of urgent cases as non-urgent, delaying timely clinical attention Same radiology label could correspond to different levels of urgency. The spectrum of urgency was not addressed in the study
Dunnmon et al., 2019 [18]	533 frontal chest radiographs	Assessed the ability of CNNs to enable automated binary classification of chest radiographs	Variety of classification CNNs	AUC of 0.96	Demonstrated the automated classification of chest radiographs as normal or abnormal	Only predicted the presence or absence of abnormality in the thoracic region Did not provide explainability
Nguyen et al., 2022 [19]	6285 frontal chest radiographs	Deployed and validated an AI-based system for detecting abnormalities on chest X-ray scans in real-world clinical settings	EfficientNet	F1 score of 0.653, accuracy of 79.6%, sensitivity of 68.6%, and specificity of 83.9%	Examined the AI performance on a clinical dataset different from the training dataset	Classified radiographs into normal or abnormal due to lack of detailed ground truth Did not check the effect of AI on radiologist diagnostic performance
Saleh et al., [20]	18,265 frontal-view chest X-ray images	Developed CNN-based DL models and compared their feasibility and performance to classify 14 chest pathologies found on chest X-rays	Variety of classification CNNs with DC-GANs	Accuracy of 67% and 62% for the best-performing model with and without augmentation, respectively	Used GAN-based techniques for data augmentation to address the lack of data for some pathologies	A different test set was used for the AI model with augmentation Test sets included images from the NIH database only
Hwang et al., [21]	1089 frontal chest X-ray images	Developed a deep learning–based algorithm that classified chest radiographs into normal and abnormal for various thoracic diseases	Variety of classification CNNs	AUC of 0.979, sensitivity of 0.979, and specificity of 0.880	AI model outperformed physicians, including thoracic radiologists. Radiologists aided with DLAD performed better than radiologists without the aid of DLAD	Validation was performed using experimentally designed data sets and included chest radiographs with only 1 target disease DLAD covered only 4 major thoracic disease categories

**Table 2 diagnostics-13-00557-t002:** Category-wise distribution of chest radiographs in the external test datasets.

Data Characteristics	No. of Chest Radiographs		
	Hospital A	Hospital B	Total
Total no. of chest radiographs	3843	633	4476
No. of radiographs with ROI in lungs	641	137	778
No. of radiographs with ROI in cardiac	114	28	142
No. of radiographs with ROI in pleura	275	50	325
No. of radiographs with at least one ROI from any category	844	163	1007
No. of radiographs with ROI in none of the above categories	2999	470	3469

**Table 3 diagnostics-13-00557-t003:** Performance of the standalone AI model on 2 external validation datasets.

Hospital	Category	AUC [95% CI]	Sensitivity [95% CI]	F1 Score [95% CI]	Specificity [95% CI]	Accuracy [95% CI]	NPV [95% CI]
A	Lungs	0.886 [0.854, 0.915]	0.846 [0.788, 0.899]	0.690 [0.636, 0.740]	0.878 [0.856, 0.900]	0.873 [0.851,0.893]	0.966 [0.952, 0.978]
	Pleura	0.867 [0.812, 0.917]	0.779 [0.672, 0.873]	0.598 [0.510, 0.677]	0.936 [0.920, 0.952]	0.925 [0.907,0.941]	0.982 [0.972, 0.991]
	Cardiac	0.919 [0.846, 0.979]	0.877 [0.739, 1.000]	0.404 [0.288, 0.508]	0.925 [0.907, 0.941]	0.923 [0.905,0.940]	0.996 [0.992, 1.000]
	Aggregate	0.910 [0.883, 0.934]	0.876 [0.829, 0.918]	0.767 [0.722, 0.807]	0.885 [0.862, 0.906]	0.883 [0.862,0.902]	0.962 [0.946, 0.975]
B	Lungs	0.902 [0.835, 0.959]	0.898 [0.781, 1.000]	0.762 [0.641, 0.857]	0.873 [0.811, 0.926]	0.878 [0.823,0.924]	0.969 [0.932, 1.000]
	Pleura	0.871 [0.734, 0.989]	0.780 [0.529, 1.000]	0.650 [0.400, 0.833]	0.947 [0.907, 0.980]	0.934 [0.892,0.975]	0.980 [0.956, 1.000]
	Cardiac	0.855 [0.641, 0.996]	0.750 [0.333, 1.000]	0.488 [0.154, 0.741]	0.939 [0.899, 0.974]	0.930 [0.886,0.968]	0.988 [0.966, 1.000]
	Aggregate	0.919 [0.864, 0.968]	0.920 [0.828, 1.000]	0.820 [0.723, 0.898]	0.887 [0.828, 0.941]	0.896 [0.848,0.943]	0.970 [0.934, 1.000]
A + B (Entire dataset)	Lungs	0.889 [0.860, 0.917]	0.855 [0.802, 0.903]	0.702 [0.651, 0.746]	0.878 [0.856, 0.899]	0.874 [0.853,0.892]	0.966 [0.953, 0.978]
	Pleura	0.867 [0.818, 0.913]	0.779 [0.686, 0.869]	0.605 [0.525, 0.682]	0.938 [0.923, 0.952]	0.926 [0.911,0.941]	0.982 [0.973, 0.990]
	Cardiac	0.906 [0.836, 0.965]	0.852 [0.720, 0.967]	0.417 [0.312, 0.514]	0.927 [0.911, 0.942]	0.924 [0.909,0.940]	0.995 [0.990, 0.999]
	Aggregate	0.912 [0.888, 0.934]	0.883 [0.840, 0.920]	0.775 [0.736, 0.811]	0.885 [0.863, 0.906]	0.885 [0.865,0.903]	0.963 [0.949, 0.975]

**Table 4 diagnostics-13-00557-t004:** Category-wise performance of the human readers in session 1 (unaided session) and session 2 (aided session) in 2 external validation datasets.

Hospital A							
Reader	Category	AUC [95% CI]		Sensitivity [95% CI]		Specificity [95% CI]	
		Unaided Session	Aided Session	Unaided Session	Aided Session	Unaided Session	Aided Session
R1	Lungs	0.776 [0.738,0.815]	0.762 [0.723, 0.802]	0.573 [0.497, 0.651]	0.537 [0.460, 0.616]	0.979 [0.969, 0.989]	0.987 [0.979, 0.994]
	Pleura	0.869 [0.819,0.917]	0.873 [0.823, 0.921]	0.767 [0.671, 0.863]	0.786 [0.685, 0.882]	0.971 [0.959, 0.981]	0.9602 [0.946, 0.973]
	Cardiac	0.809 [0.713,0.897]	0.916 [0.858, 0.956]	0.640 [0.444, 0.815]	0.929 [0.813, 1.000]	0.978 [0.968, 0.987]	0.9019 [0.884, 0.921]
	Aggregate	0.843 [0.812, 0.874]	0.861 [0.831, 0.888]	0.723 [0.662, 0.785]	0.821 [0.767, 0.871]	0.963 [0.949, 0.975]	0.900 [0.878, 0.921]
R2	Lungs	0.804 [0.766,0.842]	0.863 [0.830, 0.895]	0.632 [0.557, 0.707]	0.793 [0.731, 0.856]	0.976 [0.965, 0.986]	0.933 [0.914, 0.949]
	Pleura	0.839 [0.784,0.892]	0.872 [0.819,0.919]	0.687 [0.579, 0.795]	0.786 [0.682, 0.881]	0.989 [0.982, 0.996]	0.959 [0.945, 0.971]
	Cardiac	0.889 [0.813,0.959]	0.922 [0.867, 0.962]	0.807 [0.655, 0.950]	0.929 [0.821, 1.000]	0.971 [0.959, 0.982]	0.915 [0.897, 0.932]
	Aggregate	0.881 [0.854, 0.908]	0.901 [0.879, 0.923]	0.802 [0.749, 0.855]	0.915 [0.876, 0.950]	0.959 [0.945, 0.973]	0.888 [0.864, 0.909]
R3	Lungs	0.755 [0.714,0.794]	0.809 [0.771, 0.846]	0.589 [0.510, 0.669]	0.669 [0.593, 0.741]	0.920 [0.901, 0.938]	0.949 [0.934, 0.964]
	Pleura	0.815 [0.760,0.868]	0.874 [0.824, 0.921]	0.644 [0.535, 0.750]	0.786 [0.687, 0.878]	0.986 [0.979, 0.993]	0.962 [0.949, 0.975]
	Cardiac	0.912 [0.844,0.967]	0.8994 [0.831, 0.958]	0.886 [0.750, 1.000]	0.859 [0.722, 0.969]	0.939 [0.923, 0.954]	0.939 [0.923, 0.953]
	Aggregate	0.8352 [0.806, 0.867]	0.869 [0.839, 0.896]	0.786 [0.732, 0.844]	0.815 [0.759, 0.866]	0.884 [0.859, 0.906]	0.923 [0.905, 0.941]
**Hospital B**							
**Reader**	**Category**	**AUC [95% CI]**		**Sensitivity [95% CI]**		**Specificity [95% CI]**	
		**Unaided Session**	**Aided Session**	**Unaided Session**	**Aided Session**	**Unaided Session**	**Aided Session**
R1	Lungs	0.844 [0.762,0.919]	0.789 [0.705, 0.870]	0.723 [0.567, 0.871]	0.613 [0.447, 0.776]	0.966 [0.932, 0.992]	0.966 [0.932, 0.992]
	Pleura	0.859 [0.724,0.987]	0.901 [0.783, 0.987]	0.740 [0.467, 1.000]	0.840 [0.600, 1.00]	0.979 [0.953, 1.000]	0.962 [0.928, 0.987]
	Cardiac	0.678 [0.500,0.875]	0.929 [0.779,0.994]	0.357 [0.000, 0.750]	0.893 [0.600, 1.000]	0.998 [0.993, 1.000]	0.965 [0.933, 0.993]
	Aggregate	0.884 [0.819, 0.942]	0.885 [0.818, 0.944]	0.791 [0.667, 0.903]	0.816 [0.688, 0.927]	0.976 [0.948, 1.000]	0.953 [0.912,0.991]
R2	Lungs	0.783 [0.697,0.867]	0.835 [0.751, 0.910]	0.591 [0.419, 0.757]	0.715 [0.556, 0.862]	0.974 [0.942, 1.000]	0.954 [0.915, 0.985]
	Pleura	0.749 [0.600,0.897]	0.895 [0.764, 0.989]	0.500 [0.200, 0.800]	0.820 [0.571, 1.000]	0.998 [0.987, 1.000]	0.969 [0.938, 0.993]
	Cardiac	0.875 [0.684,0.990]	0.926 [0.773, 0.993]	0.786 [0.400, 1.000]	0.893 [0.600, 1.000]	0.964 [0.932, 0.987]	0.959 [0.924, 0.987]
	Aggregate	0.881 [0.813, 0.942]	0.923 [0.870, 0.969]	0.798 [0.667, 0.915]	0.902 [0.806, 0.977]	0.964 [0.926, 0.992]	0.945 [0.902, 0.983]
R3	Lungs	0.563 [0.499,0.632]	0.816 [0.733, 0.892]	0.183 [0.063, 0.314]	0.729 [0.571, 0.871]	0.944 [0.901, 0.977]	0.901 [0.849, 0.951]
	Pleura	0.627 [0.500,0.769]	0.894 [0.771, 0.996]	0.260 [0.000, 0.546]	0.800 [0.556, 1.000]	0.995 [0.979, 1.000]	0.988 [0.966, 1.000]
	Cardiac	0.670 [0.490,0.875]	0.923 [0.769, 0.990]	0.357 [0.000, 0.750]	0.893 [0.600, 1.000]	0.984 [0.960, 1.000]	0.954 [0.918, 0.982]
	Aggregate	0.600 [0.532, 0.676]	0.872 [0.807, 0.928]	0.258 [0.128, 0.395]	0.853 [0.738, 0.949]	0.943 [0.901, 0.982]	0.892 [0.835, 0.945]

## Data Availability

Not applicable.

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
