# Peer review of "Validation of a Deep Learning Model for Detecting Chest Pathologies from Digital Chest Radiographs"

_diagnostics, 2023, doi:10.3390/diagnostics13030557_

Round 1

Reviewer 1 Report

This is a good technical paper on detecting chest pathologies. The research output is outstanding.

However, the problem statement, contribution and proposed method are not thoroughly discussed in this paper.

I would like to suggest several improvements as below:

1. Introduction must be improved with more comprehensive reviews on chest pathologies research line. Please add comprehensive review for references [17]-[21] in the Introduction Discuss the database, problem statement, method, advantage and limitation for each reference. Conclude with a summary table of comparison. 

2. Based on the limitations of the above research line, discuss the problem statement of your research that to be solved in this paper. Please state the contribution of the paper as well.

3. The AI model (Augmento) is not comprehensively discussed in this paper. The methodology and its implementation (research design) are not clear. The motivation/rational of using this approach is also not clear. Please add these points in the paper. Has it been used in other researches? If yes, please add the discussion in the literature review part.

4. The results and discussion are excellent. The only limitation is there are no comparison results using other methods (for example from references [17]-[21]) based on your collected database. Please add this in your experiments or any reason/ limitation for not including this) in your experiments.

Author Response

This is a good technical paper on detecting chest pathologies. The research output is outstanding.

However, the problem statement, contribution and proposed method are not thoroughly discussed in this paper.

I would like to suggest several improvements as below:

Question 1: Introduction must be improved with more comprehensive reviews on chest pathologies research line. Please add comprehensive review for references [17]-[21] in the Introduction Discuss the database, problem statement, method, advantage and limitation for each reference. Conclude with a summary table of comparison. 

Answer 1: We thank the reviewer for this suggestion. We have included the comprehensive review for references [17]-[21] in a tabular form (Table 1, Page 4). We have also added a few lines in the introduction section explaining the same (Page 3, Lines 15-23)

Question 2: Based on the limitations of the above research line, discuss the problem statement of your research that to be solved in this paper. Please state the contribution of the paper as well.

Answer 2: Based on the limitations of the above papers, the problem statement and contribution are described in the introduction section (Page 3, Lines 23-31).

Question 3: The AI model (Augmento) is not comprehensively discussed in this paper. The methodology and its implementation (research design) are not clear. The motivation/rational of using this approach is also not clear. Please add these points in the paper. Has it been used in other researches? If yes, please add the discussion in the literature review part.

Answer 3: We appreciate the reviewer asking this question. We have included the details of the Augmento tool in the ‘AI model’ section (Page 7, Lines 12-15). We have also included screenshots of Augmento to bring some clarity (Figure 2). Augmento was used as an aid for radiologists to annotate scans in the MRMC study.

Question 4: The results and discussion are excellent. The only limitation is there are no comparison results using other methods (for example from references [17]-[21]) based on your collected database. Please add this in your experiments or any reason/ limitation for not including this) in your experiments.

Answer 4: We appreciate the reviewer’s comments. The results from the studies [17]-[21] are elaborated and their limitations are also mentioned in the discussion section (Page 16, Lines 10-31)

Reviewer 2 Report

Overall, the study provides strong evidence for the potential utility of the deep learning model in aiding the diagnosis of chest pathologies. However, there are some limitations to the study that should be considered.

·       For example, the study is retrospective in nature and the sample size is relatively small.

·       Additionally, the study only includes a small number of institutions and the sample of radiographs may not be generalizable to other populations.

·       Furthermore, the study should have more details about the dataset, such as the distribution of pathologies and how the data was preprocessed and augmented.

·       How the authors highlight the challenges of interpreting chest radiographs and the potential for missed detections.

·       There are some latest studies related to chest pathologies from digital images are missing so add them in the introduction or in related work section.

https://doi.org/10.3390/app12136364

ttps://doi.org/10.1155/2022/6566982

https://doi.org/10.3390/ijerph181910147

·       How the study aims to evaluate an AI system for the detection of major chest pathologies and compare its performance with experienced human readers.?

·       What are the problem statement of the research?

·       The Major contribution of the study is not clear. Must include in the introduction section.

·       How the authors mention that the AI model outperforms radiologists in accurately identifying abnormal chest radiographs and classifying them into one or more of the three different categories: lungs, pleura, and cardiac?

·       How your model is novel? Explain the novelty in more detail.

·       Can you explain more about the generalizability of the model and its performance on other datasets?

·       Are there any ethical considerations that have been taken into account in this study?

·       How the model can be improved further?

·       What are the limitations of the study?

·       Highlights some advantages of your study in the discussion section 

Author Response

Question 1: Overall, the study provides strong evidence for the potential utility of the deep learning model in aiding the diagnosis of chest pathologies. However, there are some limitations to the study that should be considered. For example, the study is retrospective in nature and the sample size is relatively small.

Answer 1: We thank the reviewer for the suggestion. We have included this limitation in the discussion section (Page 18, Lines 13-16)

Question 2: Additionally, the study only includes a small number of institutions and the sample of radiographs may not be generalizable to other populations.

Answer 2: We have addressed the limitation of generalizability in the discussion section (Page 18, Lines 13-16). As for the sample size, we do not think that the sample size used in our study is small. We used a moderate to large dataset (No. of radiographs: 4476) for our validation study. The comparison of sample size in the validation dataset of various studies is provided in Table 1.

Question 3: Furthermore, the study should have more details about the dataset, such as the distribution of pathologies and how the data was preprocessed and augmented.

Answer 3: We thank the reviewer for the suggestion. We have included the names of pathologies in the ‘Establishing Ground Truth’ section (Page 6, Lines 9-12). The details of processing and augmentations are included in the ‘AI model’ section (Page 7, Lines 4-10)

Question 4: How the authors highlight the challenges of interpreting chest radiographs and the potential for missed detections.

Answer 4: The challenges of interpreting chest radiographs using chest radiography are highlighted in the introduction section (Page 2, lines 20-31; Page 3, lines 1-3).

Question 5: There are some latest studies related to chest pathologies from digital images are missing so add them in the introduction or in related work section.

https://doi.org/10.3390/app12136364

ttps://doi.org/10.1155/2022/6566982

https://doi.org/10.3390/ijerph181910147

Answer 5: We have added the relevant studies in the introduction section (Page no., line no.)

Question 6: How the study aims to evaluate an AI system for the detection of major chest pathologies and compare its performance with experienced human readers.?

Answer 6: These points are very well covered in the results section (Page 3, lines 8-9)

Question 7: What are the problem statement of the research?

Answer 7: The problem statement of the study is described in the introduction section (Page 3, lines 23-26.)

Question 8: The Major contribution of the study is not clear. Must include in the introduction section.

Answer 8: The major contributions of the study are elaborately discussed in the discussion section (Page 17, lines 1-28) as well as in the introduction section (Page 3, lines 23-31 )

Question 9: How the authors mention that the AI model outperforms radiologists in accurately identifying abnormal chest radiographs and classifying them into one or more of the three different categories: lungs, pleura, and cardiac?

Answer 9: To ascertain whether the AI model outperforms radiologists in accurately identifying abnormal chest radiographs and classifies them into one or more of the three different categories: lungs, pleura, and cardiac, we performed a standalone AI assessment and compared it with the performance of unaided radiologists (as described in ‘Standalone performance of the AI model’ Page 9, lines 5-15; Page 10. Lines 1-3)

Question 10: How your model is novel? Explain the novelty in more detail.

Answer 10: The AI model utilized in our study classified chest radiographs into three categories, i.e., lungs, pleura, and cardiac. The strength of this approach is that the identified ROIs could be suggestive of different conditions/pathologies pertaining to these categories. This can help human readers identify the categories of the suspected abnormality and define the appropriate prognosis (Page 17, Lines 4-8)

Question 11: Can you explain more about the generalizability of the model and its performance on other datasets?

Answer 11: As reported in the results section, the model is generalizable on the datasets from two different hospitals. To determine the generalizability of the model on multiple datasets, further research including datasets from multiple geographies will be required (Page 18, Lines 13-16)

Question 12: Are there any ethical considerations that have been taken into account in this study?

Answer 12: The study was done in compliance with the Health Information Portability and Accountability Act (HIPAA). The ethical committee of both hospital 1 and hospital 2 approved the study in the institutional review board meetings. Patient information was anonymized and written informant consent was waived due to the retrospective nature of the study.

Question 13: How the model can be improved further?

Answer 13: The aim of the study is to show the improvement in reader performance when using the AI aid. Augmento was used as a platform to aid radiologists in annotating scans during the MRMC study. Therefore, discussing the limitations and improvement of Augmento is beyond the scope of this study. However, the limitations of the study design, performance, and dataset are comprehensively addressed along with their solutions in the discussion section.

Question 14: What are the limitations of the study?

Answer 14: The limitations of the study are explained in detail in the discussion section (Page 17, Lines 29-33; Page 18, lines 1-16).

Question 15: Highlights some advantages of your study in the discussion section 

Answer 15: The advantages of the study are highlighted in the discussion section (Page 17, lines 1-28).

Reviewer 3 Report

I am really grateful for reviewing this manuscript. In my opinion, this manuscript can be published once some revisions are done successfully. This study used 4476 images and Augmento (an ebsemble of deep learning models) for the detection of three chest pathologies to achieve the area under the receiver-operating-characteristic curve of 91.2% (compared to 84.2% of human experts). Firstly, I would like to ask the authors to present a more detailed explanation of Augmento in the section of Materials and Methods. Secondly, I would like to ask the authors to address in the section of Discussion how to improve Augmento for overcoming the limitations addressed. 

Author Response

Question 1: I am really grateful for reviewing this manuscript. In my opinion, this manuscript can be published once some revisions are done successfully. This study used 4476 images and Augmento (an ebsemble of deep learning models) for the detection of three chest pathologies to achieve the area under the receiver-operating-characteristic curve of 91.2% (compared to 84.2% of human experts). Firstly, I would like to ask the authors to present a more detailed explanation of Augmento in the section of Materials and Methods. Secondly, I would like to ask the authors to address in the section of Discussion how to improve Augmento for overcoming the limitations addressed. 

Answer 1: We thank the reviewer for reviewing the manuscript and providing comments. As suggested, we have incorporated a detailed explanation of Augmento in the Materials and Methods section (Page 7, lines 12-15). We have also included screenshots of Augmento tool to bring some clarity (Figure 2).

The aim of the study is to show the improvement in reader performance when using the AI aid. Augmento was used as a platform to aid radiologists in annotating scans during the MRMC study. Therefore, discussing the limitations and improvement of Augmento is beyond the scope of this study. However, the limitations of the study design, performance, and dataset are comprehensively addressed along with their solutions in the discussion section.

Round 2

Reviewer 3 Report

I am really grateful to review this manuscript. In my opinion, this manuscript can be published in current form.